# THE PRINCIPLE OF ISOMORPHISM: A THEORY OF POPULATION ACTIVITY IN GRID CELLS AND BEYOND

## ABSTRACT

Identifying the principles that determine neural population activity is paramount in the field of neuroscience. We propose the Principle of Isomorphism (PIso): population activity preserves the essential mathematical structures of the tasks it supports. Using grid cells as a model system, we show that the neural metric task is characterized by a flat Riemannian manifold, while path integration is characterized by an Abelian Lie group. We prove that each task independently constrains population activity to a toroidal topology. We further show that these perspectives are unified naturally in Euclidean space, where commutativity and flatness are intrinsically compatible and can be extended to related systems including head-direction cells and 3D grid cells. To examine how toroidal topology maps onto single-cell firing patterns, we develop a minimal network architecture that explicitly constrains population activity to toroidal manifolds. Our model robustly generates hexagonal firing fields and reveals systematic relationships between network parameters and grid spacings. Crucially, we demonstrate that conformal isometry—a commonly proposed hypothesis—alone is insufficient for hexagonal field formation. Our findings establish a direct link between computational tasks and the hexagonal–toroidal organization of grid cells, thereby providing a general framework for understanding population activity in neural systems and designing task-informed architectures in machine learning.

## 1 INTRODUCTION

Neuroscience has shifted from analyzing the tuning of individual neurons to understanding computation through population activity (Yuste, 2015; Vyas et al., 2020; Kriegeskorte & Wei, 2021; Perich et al., 2025). This transition, enabled by large-scale neural recordings (Urai et al., 2022), highlights that the functional unit of computation is the collective dynamics of neural populations (Churchland et al., 2012; Saxena & Cunningham, 2019; Langdon et al., 2023). Yet a central open question remains: how neural population activity is formed and organized.

In this work, we move beyond descriptive characterizations of neural activity to propose a principled account: population activity is determined by the mathematical structure of computational tasks it supports. More specifically, we introduce the Principle of Isomorphism, PIso, which posits that the mathematical structure inherent in a task is preserved in the structure of neural population activity.

We illustrate the power of this framework through a case study of grid cells in the mammalian entorhinal cortex, a system with well-characterized organization at both single-cell and population levels. At the single-cell level, grid cells exhibit distinctive hexagonal firing fields in physical space (Fyhn et al., 2004; Hafting et al., 2005; Sargolini et al., 2006). At the population level, cells within the same module collectively form a toroidal topology (Gardner et al., 2022). This dual-organization makes grid cells an ideal system for linking computational tasks to neural representations.

Grid cells are known to support two primary tasks: *1. path integration (PI)* (Hafting et al., 2005; Mc-Naughton et al., 2006; Burak & Fiete, 2009; Gil et al., 2018), which computes position by integrating self-motion cues, and *2. neural metric (NM)* (Moser & Moser, 2008; Ginosar et al., 2023), which provides an intrinsic metric for spatial representation. Under the principle of isomorphism, we show that each of these tasks imposes distinct structural constraints on population activity.

The seminal studies of grid cells reflect two major conceptual shifts in computational neuroscience. The first is a move from mechanistic to normative models, which explain single-cell firing patterns as solutions to optimization problems (Stachenfeld et al., 2017; Banino et al., 2018; Cueva & Wei, 2018; Sorscher et al., 2019; Whittington et al., 2020; Gao et al., 2021; Xu et al., 2022; Dorrell et al., 2022). The second is a recognition that a complete account must explain not only single-cell tuning but also the population activity (Schaeffer et al., 2022; Schøyen et al., 2023). Network studies have shown that reproducing grid-like firing at the single-cell level requires strong inductive biases (Schaeffer et al., 2022), while numerical experiments demonstrate that path integration relies on the toroidal organization of population activity rather than hexagonal fields of single cell alone (Schøyen et al., 2023). This shift has led recent computational modeling efforts to analyze both single-cell and population representations jointly (Sorscher et al., 2023; Schaeffer et al., 2023; Pettersen et al., 2024; Xu et al., 2024).

The contributions of this work are threefold (Figure 1):

- **A unifying theoretical framework**: We propose the Principle of Isomorphism, which states that neural population activity preserves the mathematical structure of tasks it supports.

- **Applications and mathematical unification**: Applying the Principle of Isomorphism, we show that Neural Metric and Path Integration correspond to flat Riemannian and Abelian Lie group structures, which independently yield toroidal population codes but naturally unify within Euclidean space. This reveals a shared representational basis that extends from grid cells to head-direction cells and 3D grid cells.

- **Architectural insights**: We design a minimal network architecture that enforces a toroidal latent manifold, enabling a systematic study of how population-level structure relates to single-cell-level hexagonal firing. With this model, we demonstrate that the previously proposed conformal isometry loss function alone is insufficient to generate hexagonal firing fields.

By applying the Principle of Isomorphism, this work provides a unified theoretical foundation for understanding grid cell representations and, more broadly, offers a general framework for linking computational function to the mathematical structure of neural population activity.

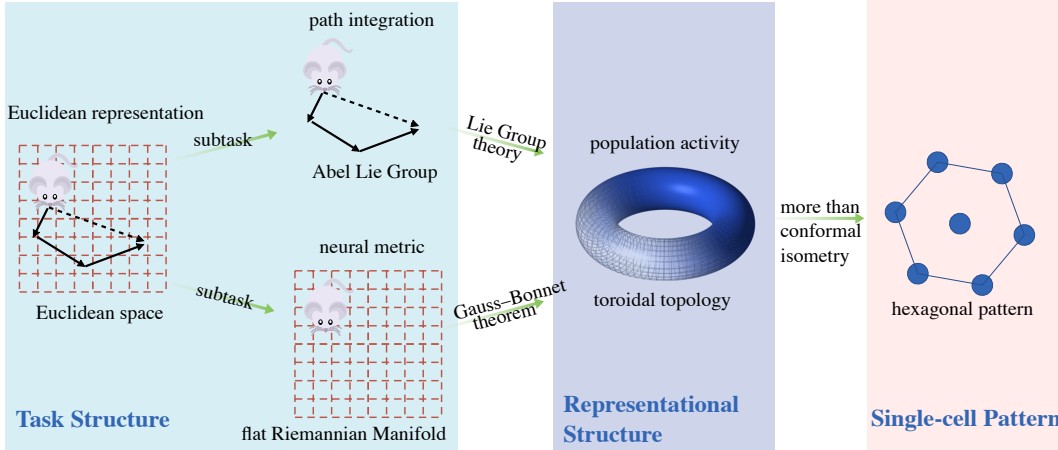

Figure 1: The principle of isomorphism applied to grid cells. Both path integration and neural metric task can be unified as the Euclidean representation task, while each independently determines the toroidal organization of population activity. Moreover, conformal isometry is not a sufficient condition for projecting toroidal population activity into hexagonal firing fields.

## 2 THE PRINCIPLE OF ISOMORPHISM (PISO)

To understand the origins of structured population activity, we introduce a theoretical framework built on two pillars: the structure of the computational task and the structure of neural population

activity. The central idea is that population activity reflects and preserves the structure of the task. We formalize this relationship through the Principle of Isomorphism.

**Task Structure**: This denotes the abstract mathematical structure embedded in a computational objective. For example, *Path Integration (PI)* is fundamentally defined by the addition of displacement vectors, an operation that forms an Abelian Lie group characterized by smoothness and commutativity. *Neural Metric (NM)* requires encoding distances and angles in physical space, which corresponds to a flat Riemannian manifold (zero curvature).

**Representational Structure**: This refers to the mathematical structure of the organization—geometric, topological, algebraic, etc.—inherent in the neural population activity. For instance, the activity of grid cells within a single module lies on a low-dimensional manifold with toroidal topology.

**Principle of Isomorphism (PIso)**: Subject to biological constraints (e.g., finite neural resources), the essential structural features of a computational task are preserved in the representational structure of the neural population. In other words, task structure and representational structure are isomorphic.

Computational tasks are defined by diverse mathematical structures, such as group algebra or metrics. Therefore, the isomorphisms that map these tasks to neural representations—such as homomorphisms or isometries—must be specifically tailored to each case, all while operating within biological limits.

PIso synthesizes ideas that have appeared across neuroscience and artificial intelligence. In neuroscience, normative theories view neural activity as evolutionarily optimized for ecological demands (Attneave, 1954; Barlow et al., 1961), early work on representational geometry emphasized preserving similarity relations (Shepard & Chipman, 1970; Edelman, 1998), and recent studies highlight encoding transformations as well as variables (Dorrell et al., 2022; Gao et al., 2021; Xu et al., 2022). In AI, related notions appear as inductive biases (Wolpert & Macready, 1997; Bengio et al., 2013), with geometric deep learning enforcing equivariance to task-defined groups (Bronstein et al., 2021), and representational alignment showing convergence across systems solving similar tasks (Sucholutsky et al., 2023).

In the following, we apply this framework to grid cells, while emphasizing that PIso is broadly applicable across neuroscience and machine learning as a general tool for analyzing population activity and guiding architecture design.

## 3    PISO IN GRID CELLS

### 3.1    NEURAL METRIC IN THE PISO FRAMEWORK

**Task Structure of NM**: The core of the NM task is that grid cells provide the brain with an intrinsic representation of physical space that preserves its geometry—specifically, distances and angles. Mathematically, such a geometry is formalized as a Riemannian manifold, defined by a metric tensor that specifies how lengths and angles are measured locally. In practice, most mammals live in a flat physical environment, so the NM task structure can be modeled as a two-dimensional flat Riemannian manifold with zero curvature everywhere.

**Representational Structure of NM**: Under the PIso framework, the representational structure of grid cells must preserve the essential features of the task structure of NM. In conjunction with basic biological constraints, this leads to the following requirements:

**Proposition 1** (Constraints of NM)**.** *The representational structure, as a manifold $M$, must satisfy:*

1. ***Two-dimensional, admitting a flat metric:*** *The manifold must admit a flat Riemannian metric, reflecting the two-dimensionality and local flatness required by NM under PIso.*

2. ***Compact:*** *Neural activity is bounded, excluding infinitely extended manifolds.*

3. ***Boundaryless:*** *Boundaries would create discontinuities that are incompatible with continuous spatial navigation Fiete et al. (2008).*

These conditions strongly restrict the global structure, such as geometry and topology, of the neural manifold $M$. We then consider the Gauss-Bonnet theorem, a fundamental result in differential geometry which links the the manifold's geometry to its topology (Needham, 2021). It states that for a compact, boundaryless 2D manifold, the total curvature is a topological invariant:

$$\int_M K \, dA = 2\pi\chi(M),$$

where $\chi(M)$ is the Euler characteristic.

**Theorem 1** (Topology implied by NM). *For a manifold that admits a flat metric (where $K = 0$ everywhere), this immediately implies $\chi(M) = 0$.*

$$\int_M K dA = \int_M 0 dA = 2\pi\chi(M) \quad \Rightarrow \quad \chi(M) = 0 \tag{1}$$

*The classification of all compact 2D surfaces reveals that only two have an Euler characteristic of zero: the torus and the Klein bottle (Hatcher, 2002). Because spatial representation must be orientable—preserving consistent notions of left/right and forward/backward—the Klein bottle is excluded, leaving the torus as the unique solution.*

The arguments above show that topology, rather than geometry or single-cell tuning, is the fundamental invariant for the NM task.

### 3.2 PATH INTEGRATION IN THE PISO FRAMEWORK

The Path Integration (PI) of grid cells has been extensively studied (Fiete et al., 2008; Sreenivasan & Fiete, 2011; Whittington et al., 2020; Gao et al., 2021; Xu et al., 2022). Restating this within our PIso framework provides a unifying perspective:

- **Task Structure**: Path integration is defined by the summation of displacement vectors, which satisfies the algebraic structure of the Abelian Lie group $(\mathbb{R}^2, +)$.

- **Isomorphic Constraint**: The representational structure must therefore be a connected, two-dimensional Abelian Lie group.

- **Biological Constraint**: Neural activity is bounded, so the Lie group must be compact.

According to the theory of compact Lie group, the only possible topology of this Lie group is torus (Dwyer & Wilkerson, 1998). Thus, from the PI perspective, the toroidal topology emerges directly from the requirement that the neural population acts as a commutative group for integrating displacements. This confirms that, as in NM, topology is an invariant underlying PI.

### 3.3 UNIFYING NM AND PI: EUCLIDEAN REPRESENTATION

Having shown that NM and PI, each independently constrain the patterns of grid cell neural population activity to a toroidal structure, we now turn to their formal relationship. Within Euclidean space, these two tasks can be unified through the intrinsic compatibility between flatness (Riemannian structure) and commutativity (group structure). This perspective reveals Euclidean representation as the deeper task structure that subsumes both NM and PI under the PIso framework (Figure 1).

Specifically, this unifying can be understood at both the task-structure as well as the representational level. At the task-structure level, NM's Riemannian structure and PI's group structure are jointly realized in two-dimensional Euclidean space. As a vector space, Euclidean space supports the Abelian addition required for PI, and as an inner product space it admits the flat Riemannian metric required for NM. Therefore, group actions preserve the inner product structure within Euclidean space, ensuring that translations and rotations are isometries. This yields two fundamental symmetries: **Homogeneity**, no privileged locations—geometric properties are invariant across all

positions. **Isotropy**, no privileged directions—geometric properties are invariant across all orientations. Together, these symmetries reflect the structure of Euclidean representation, within which NM and PI emerge as complementary facets of a single unified task.

At the representational level, the only compact manifolds compatible with Abelian group and flat Riemannian structures in any dimension are tori. Thus, the unification naturally extends beyond 2D grid cells to related spatial navigation systems (Rank, 1984; Ginosar et al., 2021; Grieves et al., 2021). Head-Direction Cells can be viewed as computing a one-dimensional version of PI and NM, with an intrinsically circular task structure rather than an unbounded Euclidean line. 3D grid cells would be expected to form a 3D torus. However, the lack of robust periodicity in recordings suggests either additional neural mechanisms that distort this canonical structure, or fundamentally different computational demands in volumetric navigation (Grieves et al., 2021).

This unification also constrains plausible physiological mechanisms underlying grid cell formation. It explains why successful Continuous Attractor Neural Network models rely on synaptic connectivity that enforces translational invariance—a biological implementation of the homogeneity required by the Euclidean task (Skaggs et al., 1994; Zhang, 1996; Burak & Fiete, 2009).

## 4 EXPERIMENTS

The above PIso framework demonstrate that the representational topology—not single-cell tuning—is functional invariant of spatial computation. Many of previous network simulations have consistently revealed toroidal manifolds but differ in their single-cell projections: some models produce band-like firing fields, while others yield hexagonal patterns (Schøyen et al., 2023; Pettersen et al., 2024). These observations have been interpreted as evidence that bands and hexagons firing fields support distinct computations (PI vs. NM). In contrast, our framework suggests that both computations arise from the shared toroidal substrate. Topology determines whether a task is possible at all, whereas single-cell firing patterns influence performance, providing alternative readout strategies with different efficiencies.

This raises a central question: why does the entorhinal cortex, when projecting toroidal population activity onto individual cells, predominantly produce hexagonal firing fields? To address this, we developed the Topo-Constrained Network (TopoCN), a computational framework that explicitly constrains population activity to toroidal manifolds while systematically explores single-cell firing properties.

### 4.1 ARCHITECTURE OF THE TOPO-CONSTRAINED NETWORK (TOPOCN)

We designed a minimal feedforward, self-supervised network with a conformal isometry (CI) loss (Figure 2a). Self-supervised learning has been used in grid-cell modeling (Schaeffer et al., 2023), and CI loss was introduced to preserve local metrics (Gao et al., 2021; Xu et al., 2022). A similar architecture was explored by Pettersen et al. (2024) for NM task. Our approach differs in two key ways: (1) we explicitly enforce toroidal population activity (Figure 2b, c), and (2) we replace capacity-related losses (Schaeffer et al., 2023; Pettersen et al., 2024) with control of torus size, which we systematically vary. Thus, our model serves not to replicate NM, but to test how toroidal topology maps onto single-cell firing patterns. More detailed explanations, including an RNN-based PI extension of TopoCN and a comparison between torus-size control and capacity-based regularization, can be found in Appendix B and Appendix C.

Consider a square-shaped 2D arena (see Appendix A for details). Each location $(x, y)$ is mapped to a 4D toroidal embedding $\mathbf{n} \in \mathbb{R}^4$:

$$
\begin{aligned}
n_1 &= R\cos(k_1 x + k_2 y), \\
n_2 &= R\sin(k_1 x + k_2 y), \\
n_3 &= r\cos(k_3 x + k_4 y), \\
n_4 &= r\sin(k_3 x + k_4 y),
\end{aligned}
\tag{2}
$$

where $R, r, k_i$ $(i = 1, \ldots, 4)$ are learnable parameters, with $R$ and $r$ setting the major and minor radii of the torus and $k_i$ specifying the linear map from physical coordinates to torus angles. This guarantees that the network input already lies on a toroidal manifold, from which downstream layers can perform diverse transformations.

The embedding $\mathbf{n}$ is then passed to a multi-layer perceptron $MLP : \mathbb{R}^4 \to \mathbb{R}^G$, whose output $\mathbf{g} = MLP(\mathbf{n})$ represents the activities of a population of $G$ grid cells. We impose two generic biological constraints:

$$\|\mathbf{g}\|^2 = 1, \quad g_i \geq 0 \ \ \forall i \in \{1, \ldots, G\}, \tag{3}$$

where the constant $\ell_2$-norm reflects is supported by empirical observation in the experiment (Xu et al., 2024), and non-negativity reflects firing-rate constraints.

To probe how toroidal structure shapes single-cell firing patterns, we systematically vary two hyperparameters: (1) *Scaling factor $\rho$*, which controls the mapping between physical space and neural representation; (2) *Torus size $s$*, which quantifies the compactness of the toroidal manifold:

$$s = 1 - \left\| \frac{1}{m} \sum_{k=1}^{m} \mathbf{g}^{(k)} \right\|_2^2, \tag{4}$$

where $\mathbf{g}^{(k)}$ is the output representation of the $k$-th randomly sampled location in physical space. This metric ranges from 0 to 1, with larger values indicating more distributed toroidal representations.

The network is trained with a composite objective that combines a conformal isometry term with a torus size regularization:

$$\mathcal{L} = \mathbb{E}\left[ \left( \|\Delta \mathbf{g}\| - \rho \|\Delta \mathbf{x}\| \right)^2 \exp\left( -\frac{\|\Delta \mathbf{x}\|^2}{2\sigma^2} \right) \right] + \lambda(s - s_0)^2, \tag{5}$$

with the constraint $\|\mathbf{g}\|^2 = 1$. Here $\Delta \mathbf{x}$ and $\Delta \mathbf{g}$ denote displacement in physical and the corresponding change of representation vector in the neural space, $\sigma$ controls the locality of conformal isometry, $\lambda = 2$ and $s_0 \in [0, 1]$ sets the target torus size. In all the numerical results reported in the main text, $G = 32$ was used.

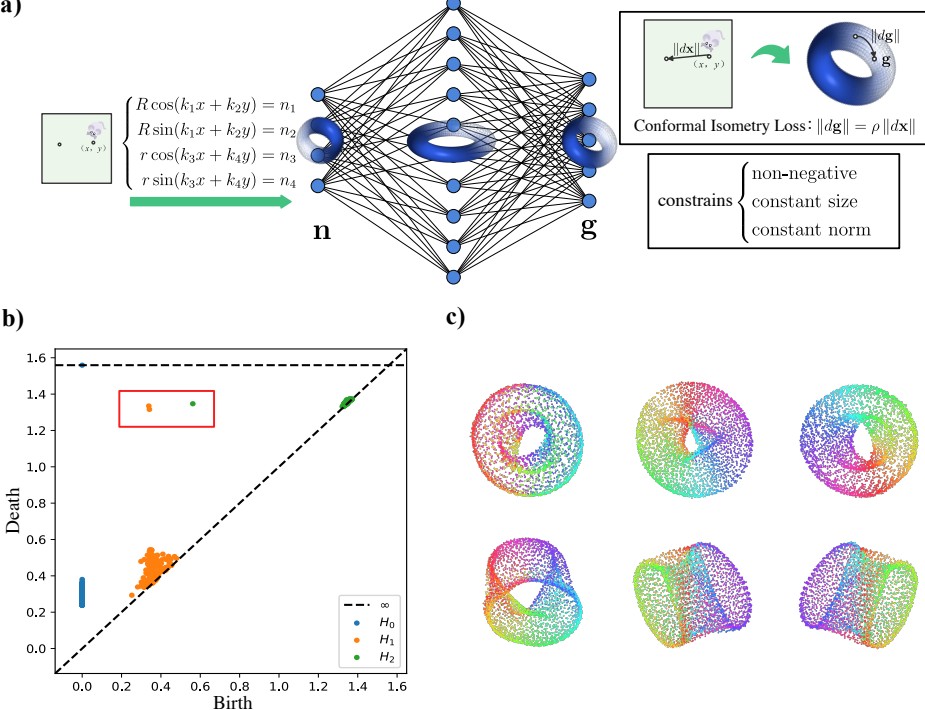

Figure 2: (a). Our minimal network model to explore how to generate hexagonal firing field from torus population topology. (b) Persistent homology analysis showing the torus structure. (c) Visualization of the torus population topology from numerical simulation (using PCA followed by UMAP for dimensionality reduction).

This experimental framework enables us to test whether hexagonal firing patterns emerge naturally from toroidal constraints or require additional constraints.

## 4.2 EXPERIMENTAL RESULTS

### 4.2.1 SPONTANEOUS EMERGENCE OF HEXAGONAL FIELDS

Within our minimal architecture and without additional constraints, toroidal population activity naturally projects to hexagonal grid-like firing fields. These fields exhibit uniform spacing and orientation across neurons, consistent with experimental observations of single grid modules (Figure 3) (see Appendix E for more firing fields). At large torus sizes, spacing variations frequently emerge, suggestive of spontaneous subdivision into multiple modules—an adaptive mechanism to increase representational capacity (Schaeffer et al., 2023).

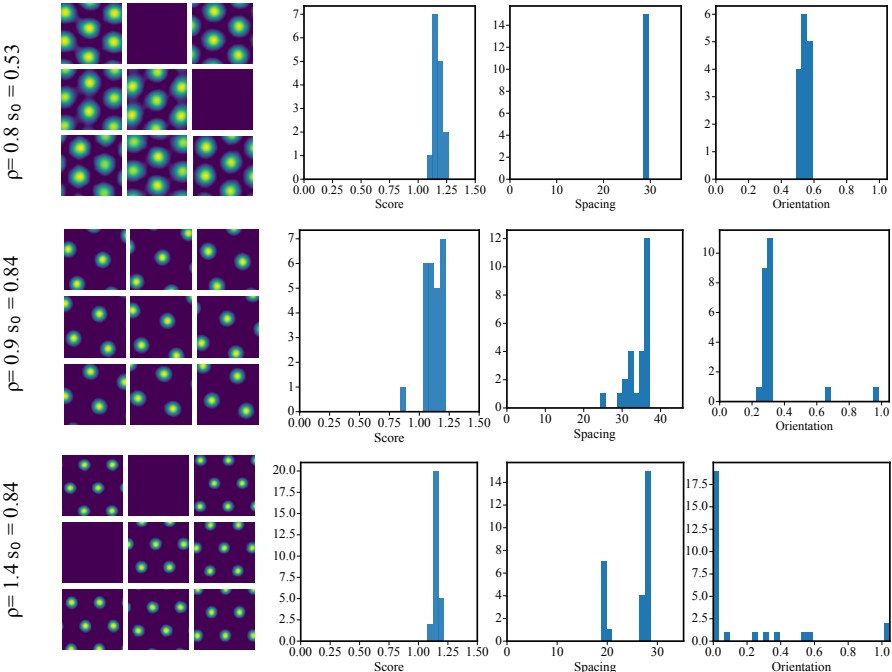

Figure 3: Examples of grid-cell firing fields under different hyperparameters $\rho$ and $s_0$ (left). Distribution of grid score, spacing, and orientation (right) shows narrow clustering, consistent with a single grid module.

### 4.2.2 ROBUSTNESS TO NOISE

To test resilience of the single-neuron firing patterns to noise, we inject Gaussian noise into the output neural activities and quantify (1) the proportion of active neurons and (2) the fraction of active neurons that exhibit hexagonal firing fields. As noise increased, the network recruits additional neurons to enhance robustness, while hexagonal patterns persist across a wide range of noise levels (Figure 4a). This shows that toroidal-to-hexagonal projection is robust to realistic neural variability.

### 4.2.3 LIMITS OF CONFORMAL ISOMETRY

Systematically varying torus size $s_0$ while holding mapping scale $\rho$ constant reveals a striking relationship between topology and firing patterns (Figure 4b): (1) the proportion of active neurons increase with torus size, consistent with coverage of larger manifold areas; (2) the probability of grid-like cells peaks at intermediate values of $s_0$; (3) excessively large $s_0$ produce square-like rather than hexagonal fields, while small $s_0$ generate diffuse, non-localized responses.

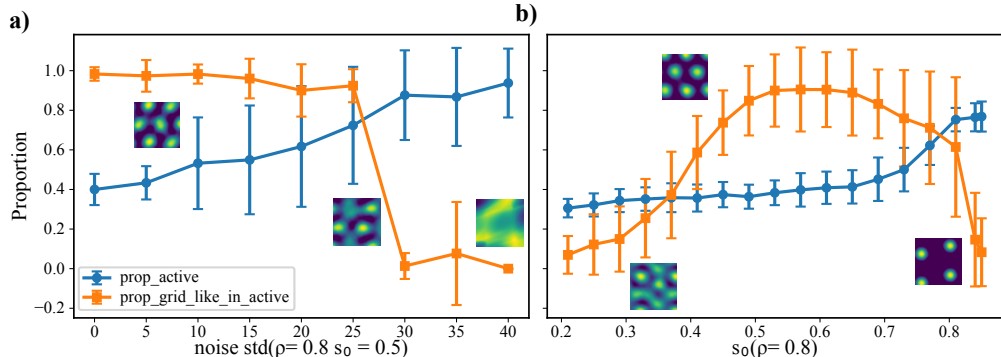

Figure 4: Proportion of active cells and grid cells as a function of (a) noise level and (b) torus size parameter $s_0$. Hexagonal firing fields are robust to moderate noise, but only emerge within an intermediate range of torus sizes.

Xu et al. (2024) showed that conformal isometry can promote toroidal-to-hexagonal projection by maximally flattening the torus. Our experiments reveal that this outcome occurs only within a restricted range of torus sizes. Thus, CI is not a sufficient condition for hexagonal fields formation, pointing to the existence of deeper geometric or optimization mechanisms yet to be uncovered.

### 4.2.4 GRID FIELD SPACING AS A FUNCTION OF $\rho$ AND $s_0$

The grid spacing—the physical distance between adjacent firing fields—reflects the distance corresponding to one complete traversal around the torus in a given direction. Since the scaling factor $\rho$ controls the mapping between physical displacement and neural-space displacement, we predict that, when the torus size $s_0$ is held constant, grid spacing should be inversely proportional to $\rho$. This prediction is clearly confirmed by our simulations (Fig. 5a) (see Appendix D for details). Similar relationship is also reported in Xu et al. (2024), where RNNs were trained to perform PI with a slightly different objective function.

Furthermore, we investigated the influence of the torus size $s_0$ while holding the scaling factor $\rho$ constant. We observe that grid spacing increases monotonically with torus size (Figure 5 b). This relationship suggests that larger toroidal manifolds naturally accommodate broader spatial representations, with individual grid fields covering correspondingly larger regions of physical space. These findings demonstrate that both parameters $\rho$ and $s_0$, exert independent and predictable control over grid spacing.

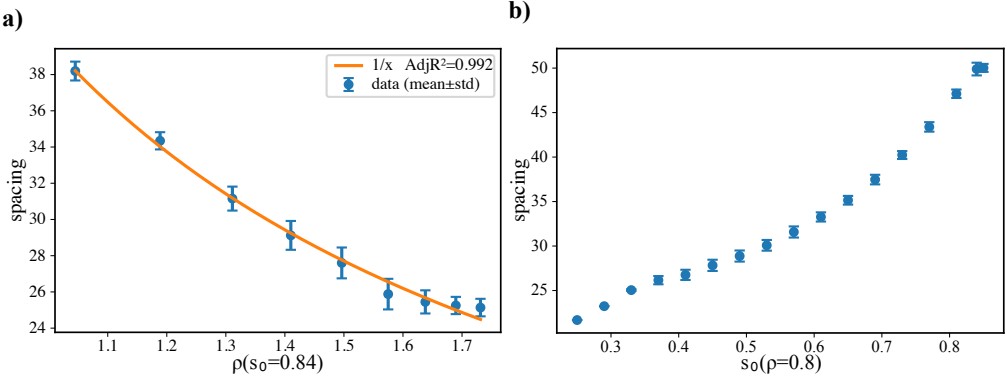

Figure 5: Grid spacing is proportional to $1/\rho$ and monotonically increases with the torus size parameter $s_0$.

## 5 RELATION TO PAST WORKS

The studies by Gao et al. (2021); Xu et al. (2022) were among the first to analyze grid cells from a geometric perspective and to propose the conformal isometry hypothesis. This hypothesis posits that distances on the neural manifold locally match physical distances up to a scaling factor, thereby constraining both the geometry and the mapping. However, its strict requirements appear inconsistent with experimental observations of distorted grid patterns (Krupic et al., 2015; 2018; Stensola et al., 2015). In contrast, our framework for NM task only requires the existence of a flat metric, which constrains the population's topology (to a torus) but allows for flexibility in its precise geometry. This topological focus accommodates the observed distortions, as the toroidal structure remains invariant even when the geometric embedding changes (Gardner et al., 2022). In cases where distortions are minimal, population activity may indeed approximate conformal isometry (Xu et al., 2024), enabling simple linear decoding (Pouget et al., 2000; Burak & Fiete, 2009; Sreenivasan & Fiete, 2011; Stemmler et al., 2015). Our perspective generalizes this, suggesting that for any toroidal manifold—even a geometrically distorted one—a decoder can recover the flat geometry, though it may need to be nonlinear. This contrasts with non-toroidal manifolds, for which such a recovery is fundamentally impossible.

Beyond geometry and NM, another important line of work has focused on PI. Prior studies have shown that grid cell activity preserves the algebra of displacement during PI (Fiete et al., 2008; Sreenivasan & Fiete, 2011), as well as the commutativity of displacement summation (Whittington et al., 2020). Notably, Gao et al. (2021); Xu et al. (2022) highlighted the 2D additive group structure of displacements $(\mathbb{R}^2, +)$, interpreting PI as a computational function that naturally gives rise to a group representation of this structure. Under the requirement that such a representation be both compact and connected, the neural population activity must therefore take the form of a torus. Our discussion in §3.2 can be viewed as a reinterpretation of these prior ideas within the unifying PIso framework.

## 6 DISCUSSION

We introduced the *principle of isomorphism* as a general framework for linking computational tasks to neural population activity. Applying this framework to grid cells, we showed that PI and neural metric NM emerge as complementary subcomponents of a unified Euclidean representational task, each independently constraining population activity to a toroidal topology. This establishes topology as the functional invariant of spatial computation. Through numerical experiments, we systematically explored the relationship between toroidal population activity and the emergence of hexagonal single-cell firing fields. These analyses revealed that conformal isometry loss alone is insufficient to produce hexagonal patterns.

Our framework generates clear testable predictions. A critical test will be to experimentally probe the population topology of 3D grid cells to determine if it conforms to a 3-torus (as predicted by a straightforward extension of 2D systems ) or a different structure, which would indicate fundamentally different computational constraints.

Our work leaves important theoretical questions. While we have established the necessity of a torus for the computational tasks, a complete formal derivation of why its projection yields hexagonal firing fields so reliably remains a challenge. Furthermore, we see significant promise in extending PIso beyond spatial cognition to serve as a general tool for analyzing population codes across neural systems and artificial networks.

Moreover, PIso suggests a new design principle for artificial intelligence: the use of a *Topology Prior*. For tasks with Euclidean structure, constraining latent representations to a torus provides a natural inductive bias. More generally, when tasks involve non-Euclidean geometries (e.g., spherical or hyperbolic), the population activity should be constrained to manifolds with the corresponding topology, as required by mathematical results such as the Gauss–Bonnet theorem. Embedding topology directly into network design thus offers a path toward more efficient, robust, and interpretable neural architectures.

## ACKNOWLEDGMENTS

We acknowledge the use of large language models (e.g., ChatGPT) for assistance with language editing and improving readability. All conceptual, experimental, and theoretical contributions are the authors' own.

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

## A    NUMERICAL SIMULATION SETUP

Our implementation was partly adapted from the publicly available code of Pettersen et al. (2024). We thank the authors for releasing their code. On top of their framework, we explicitly enforced toroidal population activity, introduced the torus-size regularization and systematically explored its interaction with conformal isometry loss, as described in the main text.

**Environment and data generation.** We modeled the environment as a two-dimensional square plane. During training, batches of positions were uniformly sampled from $[-2\pi, 2\pi]^2$; during evaluation, inputs were taken from a regular $64 \times 64$ meshgrid covering the same domain.

**Network details.** The input $(x, y)$ was first encoded by a custom `ToriActivation` layer, which mapped the environment to a torus, and then passed through a multilayer perceptron with ReLU activations. Gaussian noise was added to the last layer for regularization, and the final activity vector was normalized to enforce the $\ell_2$ constraint ($\|\mathbf{g}\|_2 = 1$). Non-negativity was guaranteed by the ReLU nonlinearity. The network therefore implemented a minimal feedforward mapping from 2D coordinates to a $d$-dimensional neural population vector.

**Training procedure.** We trained the model using the Adam optimizer with learning rate $10^{-3}$, batch size $64$, and $50{,}000$ training epochs.

**Grid analysis.** To quantify the learned representations, we computed grid score, orientation and spacing of unit rate maps following the procedure of Pettersen et al. (2024). Rate maps ($64 \times 64$) were smoothed with a Gaussian kernel. Grid score was defined as the difference between autocorrelogram correlations at $60°$ and $30°$; orientation as the smallest angle between the horizontal and the six innermost peaks (excluding the origin); spacing as the average distance to these peaks; and phase as the displacement of the nearest peak from the origin.

**Hardware.** All the models were trained on NVIDIA GeForce RTX 4090 (24 GB). For a single model, training time was less than 15 minutes.

## B    RNN-BASED PATH INTEGRATION

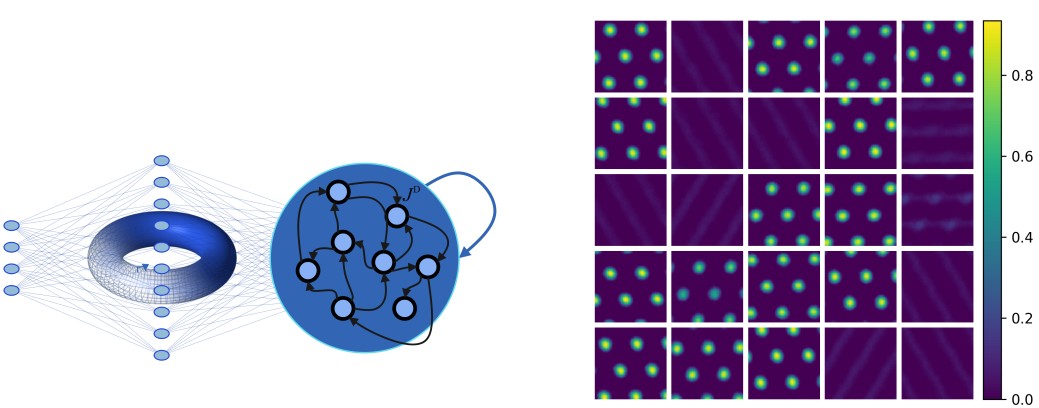

(a) RNN-based TopoCN architecture.          (b) Firing fields of RNN-based TopoCN.

Figure 6: The architecture and firing fields.

To assess robustness of the hexagonal firing patterns, we coupled the Topo-Constrained Network (TopoCN) with a recurrent neural network that performs path integration (Pettersen et al., 2024; Xu et al., 2024). Using the same environment and trajectories sampled as in the main experiments, TopoCN only encodes the starting location on a toroidal manifold, while the RNN receives stepwise

displacements and iteratively updates the location representation from this initialization. Training employed the same conformal isometry (CI) loss as in the main text. Remarkably, hexagonal firing fields emerged reliably in this RNN–PI setting without any explicit torus-size regularization or capacity penalties (Figure 6). In the next subsection we explain this observation by analyzing how the dynamics and CI constraint drive the torus size toward an intrinsic stable range.

## C TORUS SIZE VERSUS CAPACITY REGULARIZATION

We compared three training regimes on TopoCN: (i) no torus-size regularization; (ii) replacing the size term (the second term in equation 5) with an $L^1$ **capacity regularization** (Pettersen et al., 2024); and (iii) replacing the size term with an $L^2$ **capacity regularization** (Schaeffer et al., 2023). The outcomes were consistent across runs (Figure 7):

- **No regularization**: grid spacing became *non-uniform*, and fields tended to be more *square-like* than hexagonal;
- $L^1$: fields were generally more *diffuse*;
- $L^2$: clear *hexagonal* grid-like fields emerged.

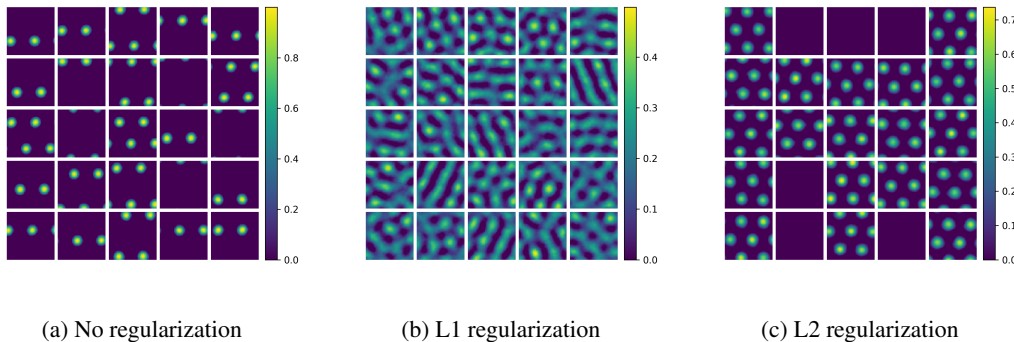

(a) No regularization         (b) L1 regularization         (c) L2 regularization

Figure 7: Firing fields of TopoCN trained with different regularization schemes.

Formally, the two types of capacity regularization can be written as follows. The $L^2$ version (Schaeffer et al., 2023) penalizes the squared norm of the mean population activity:

$$\mathcal{L}_{\text{cap}}^{(2)} \;=\; -\left\| \frac{1}{n} \sum_{k=1}^{n} \mathbf{g}^{(k)} \right\|_2^2, \tag{6}$$

which encourages broad, uniform representations on the hypersphere and is closely related to our torus size measure in equation 4.

The $L^1$ capacity constraint promotes maximally correlated activity across neurons, effectively pushing the population activity toward the diagonal of the state space where all units are coactive:

$$\mathcal{L}_{\text{cap}}^{(1)} \;=\; -\sum_{i} g_i, \tag{7}$$

Closer inspection revealed that the apparent success of the $L^2$ capacity regularization is not a general property, but rather a coincidence under our experimental settings: in this case, $L^2$ happened to steer the torus size $s$ into the regime where hexagonal fields emerge. A similar situation arises in the RNN experiments: under our settings, even *without* explicit regularization, the interaction between CI and recurrent dynamics *self-organizes* the torus size into the hexagon-supporting regime.

These observations point to torus size—not capacity—as the true factor underlying hexagonal grid formation. To avoid relying on such coincidences, we replace capacity penalties with a more transparent experimental knob: **directly controlling and systematically exploring the torus size** to probe how toroidal population topology projects onto single-cell firing patterns.

# D  PRACTICAL HANDLING OF $\rho$

As shown in the main text, grid spacing decreases with the scale factor $\rho$ (roughly as $1/\rho$) when torus size $s_0$ is fixed, and increases with $s_0$ when $\rho$ is fixed (Fig. 5).

In practice, however, the loss function uses chord length in neural space rather than true geodesic length. This introduces a small mismatch: the actual scale realized by the trained network, denoted $\rho_{\text{true}}$, may differ from the nominal $\rho$ set during training. As a result, spacing measured from neural activity is determined by $\rho_{\text{true}}$, not the nominal value.

To deal with this, we distinguish two cases:

**Varying $\rho$ with fixed $s_0$:** We compare spacing as a function of the *nominal* $\rho$ and the *measured* $\rho_{\text{true}}$, and fit the latter with the predicted hyperbolic law sp $\propto 1/\rho$. This shows that the effective scale realized by the network follows the theoretical prediction (Fig. 8a).

**Varying $s_0$ with fixed $\rho$:** Even if $\rho$ is fixed nominally, small deviations in $\rho_{\text{true}}$ can still affect spacing. To remove this confound, we normalize spacing to a common reference scale $\rho_{\text{ref}}$ using

$$\text{sp}_{\text{corr}} = \text{sp} \times \frac{\rho_{\text{true}}}{\rho_{\text{ref}}}, \tag{8}$$

which follows from the proportionality sp $\propto 1/\rho$. We therefore report both the raw spacing–vs–size curve and the $\rho$-corrected version for clarity (Fig. 8b).

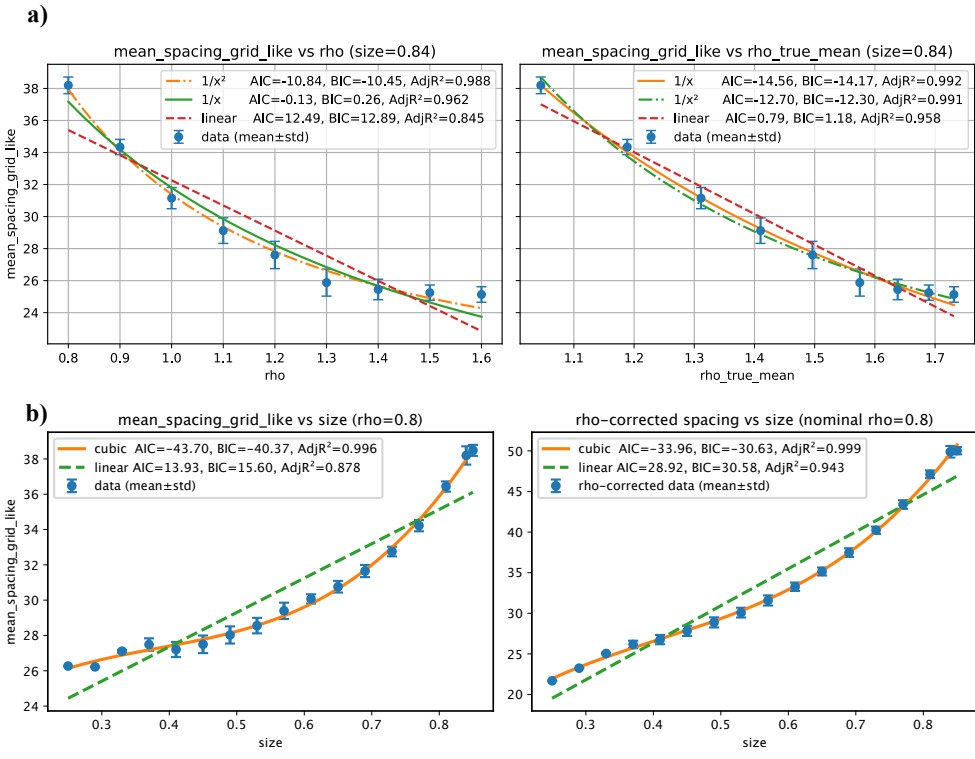

Figure 8: (a) Comparison of spacing as a function of nominal versus effective $\rho$. (b) Spacing–size relationship before and after correcting for $\rho$.

Interestingly, after correcting for the residual influence of $\rho$, the relationship between spacing and torus size $s_0$ follows a clear cubic law. Unlike the inverse dependence on $\rho$, which can be derived analytically, we do not yet have a theoretical explanation for this cubic dependence. Nevertheless, the fit is highly robust across simulations, suggesting that it reflects a genuine structural property of

the model rather than a numerical artifact. We leave a full theoretical analysis of this phenomenon to future work.

# E  ADDITIONAL FIRING FIELD EXAMPLES

To complement the main text, Figure 9 presents additional examples of firing fields generated by TopoCN under different training regimes. These visualizations illustrate the variability across runs while consistently demonstrating the emergence of hexagonal structure under appropriate torus-size constraints.

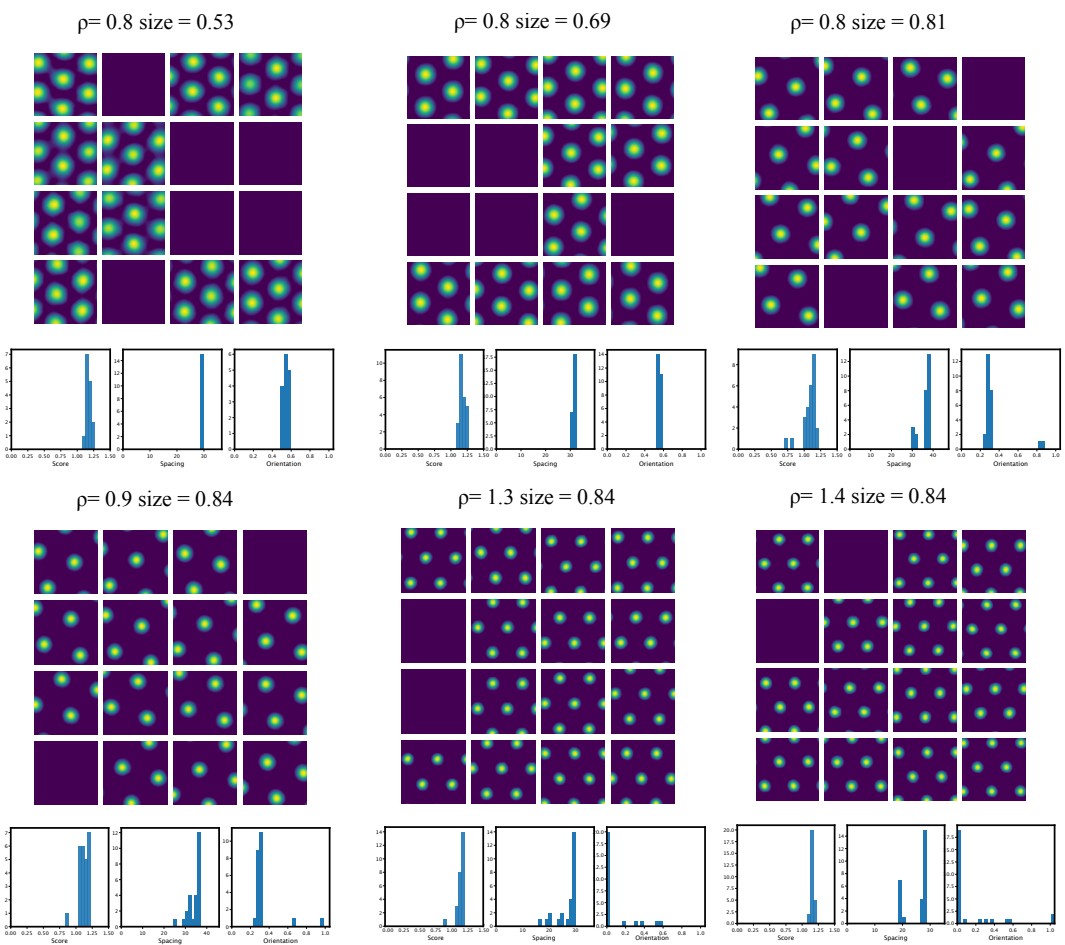

Figure 9: Additional Firing Field Examples.

