# OpenReview forum: "The Principle of Isomorphism: A Theory of Population Activity in Grid Cells and Beyond"
_ICLR.cc/2026/Conference — ICLR 2026 Conference Withdrawn Submission_

### Official Review · Reviewer_foPi · 2025-10-24

**Soundness:** 4
**Presentation:** 3
**Contribution:** 1
**Rating:** 4
**Confidence:** 4

**Summary:**

This paper considers the classic question, why do we have grid cells?

They formalise the impact of two ideas, path-integration and metric-embedding, and use them both to arrive at the idea of a torus. Then they constrain activity to lie on a torus and optimise a mapping of the torus to achieve other losses, finding a nice combination that produces hexagonal grid cells. They measure the effect of parameters on these conclusions.

They suggest these ideas generalise.

(When I reference line numbers, it refers to the numbers on the left hand side of the pages)

**Strengths:**

- The (relatively abstract (to me!)) mathematical ideas were well explained
- There were many links to existing works that were well explained
- The core ideas are good
- I found the goals of the paper interesting
- The experiments seem well performed
- I really liked the torus + neural network formulation of your theory, it was neat

**Weaknesses:**

I have three main concerns with this paper that lead me to recommend weak rejection.
- First, while the paper was nicely written and had some clean ideas, it seemed that the novel contribution of the paper was relatively limited. Rather, a lot of the work was reframing of existing ideas. Maybe that’s okay, because the framing was nice, but I’m not sure how much I learnt from the paper relative to existing work.
- Second, the sloppy and grandiose logic (in the text, the maths/simulation logic seemed sound) gave me repeated aneurisms. This I suspect could be fixed by a good rebuttal (and substantial rewriting).
- It is framed as path-integration + metric-embedding -> grid cells, but it seemed that only one was necessary, which seems like a fundamentally different conclusion than the authors draw, and the lapses between intrinsic and ambient metric were confusing and common,

An answer or edits responding to points 2 and 3 could probably move my vote to a weak accept. Evidence of a more substantial contribution could raise it further, but, unfortunately, I find such a change hard to imagine.

I will now outline why I hold each of these beliefs.

I'll begin with the third point, PI + NM. it is stated that grid cells are 'known to support two tasks: path-integration ... and neural metric'. I agree that the evidence for PI is strong, but I don't get why I should think grid cells are also doing NM? Mechanistic models that rely only on PI generate grid cells (Burak & Fiete, 2009), normative theories without NM generate grid cells (Whittington et al. 2020, Dorrell et al. 2023), and the two papers the authors cite as evidence (Moser & Moser, 2008; Ginosar et al. 2023) are reviews that happen to have the word metric in the title. However, translating Moser & Moser 2008 into computational language, they seem to suggest PI is the computational goal of grid cells, but never argue explicitly for NM, since the grid metric does distort in the real world in different environments (Stensola et al. 2012). In that case, why should I think NM is a reasonable criterion for grid cells?

This point becomes especially important when a lot of the authors description of the ideas suggest they view PI and NM as two equivalent ways to get to a torus. I think it is the authors’ opinion, judging from sentences such as ‘Having shown that NM and PI, [why a comma?] each independently constrain the patterns of grid cell neural population activity to a toroidal structure’. If they are both independently performing the same job, why do you need both? How could we distinguish a circuit that did one not the other in experiments? If you never can, are they really different ideas or two ways of thinking about the same idea? Perhaps I am misunderstanding here. This seems especially likely given that I found section 3.3 very puzzling - it seemed to say that 2D euclidean space allows both NM and PI and is a fusing of the two ‘NM’s riemannian structure and PI’s group structure are jointly realized in two-dimensional eucoidean space’, but… of course it is, because you derived NM and PI from the constraint of navigating around 2D space…??? Or, in other words, NM and PI are both created to reflect the structure of 2D space, and 2D space satisfies both….

Putting the previous point to one side, I think the authors’ dissection of which parts of their model were NM was hard for me to grasp, and potentially inaccurate. First, they say that ‘topology, rather than geometry or single-cell tuning, is the fundamental invariant for the NM task’; but they explicitly say that, for them, NM means having a flat riemmanian metric. Is a metric not an explicitly geometric object? They reached the topology through geometry, why then is it reasonable to say topology is fundamental?

Then later they use a conformal isometry loss (eqn. 5), is that not an additional NM-like idea? My understanding is that the intrinsic metric is flat (hence torus) but the ambient neural-space metric is not, and that’s what the conformal isometry loss constrains. As such, the authors seem to think that both the intrinsic and ambient metric are meaningful, why? If the authors do like conformal isometry, why do they think the grid lattice shears in square rooms (Stensola et al. 2012)?

A further way this confusion impacts the paper: at the start of section 4 the authors argue that their findings help to illuminate the debate about bands and hexagons, and NM vs. PI in schoyen et al. 2023 & Pettersen et al. 2024. But, this is a slip from intrinsic metric (NM) to ambient euclidean metric. The NM Pettersen and Schoyen are talking about is the ambient version, exactly as in the conformal isometry loss in eqn 5. This is different, I think, from what the authors think NM is, so why is there any useful link to draw between the two? There is still the puzzle of why it seems like in Pettersen/Schoyen grid cells do ambient-NM (conformal isometry) while band cells do PI that the authors discussion doesn't clear up.

Now I’ll turn to the first point, the slightly limited contribution of the paper. The result of the main data figure of this paper (fig 3) is that a torus, derived from either NM (a new contribution) or PI (an old contribution), + a loss that is morally identical to that used by, for example, Pettersen et al. 2024 produces one module of axis-aligned hexagonal grid cells. Great, that is a novel result. The authors are also right to point out that conformal isometry/eqn 5 alone doesn’t generate grid cells, and further discuss how, even when Pettersen et al. ask their RNN to path-integrate, they get band cells. Given that the authors argue PI is equivalent to a torus, do the authors understand this discrepancy? But how do they view their contribution relative to other models that generate multiple modules of axis-aligned grid cells, often using similar ideas (Dorrell et al. 2023, Schaeffer et al. 2023)?

The authors take one route towards a bigger contribution, they frame their work as much more general. While the ideas and text were nice to read and I broadly agree with what the authors say, I wasn’t sure there was much new science in there, making it a similarly limited contribution. Everyone thinks it would be great if the ideas about grid cells generalise to other tasks - continuous attractor networks are routinely used for angles and spheres, as well as 2D space, people have long thought about grid cells in 3D space, or on sloping surfaces. Giving this existing evidence of ideas generalising beyond 2D space, what concrete ideas should I be taking from the authors about how to generalise? Some evidence of how their ideas are new and let them generalise would be needed to justify the claim that this is a usefully general idea.

Another route to a larger contribution would be explaining how the current framework for grid cells helps us understand something that others couldn’t, but I see no evidence of this.

As such, the concrete scientific contribution of the paper, that NM = torus, and that torus + eqn 5 leads to grid cells, is interesting and well done, I liked it. But it is fairly limited, which caps the score I am likely to give this paper.


Finally, grandiose and/or sloppy writing:
- Line 040: the phrase "we move beyond descriptive" suggested that the field had only done descriptive work before, which is, of course, factually not true. There have been tens of normative/mechanistic models of grid cells before, let alone the rest of the neuroscience.
- Line 54: framing the move from mechanistic to normative models as a ‘major conceptual shift’ is nuts. Both are useful for different things, and should, and always have, existed alongside each other: the earliest grid cell work contains both normative theories (Mathis et al. 2011) and mechanistic (Burak & Fiete, 2009), and this pairing continues.
- Line 58: A similarly confusing claim about the move from single cell to population, both of the two papers above are clearly about populations of neurons. What exactly is different?
- Line 227: What claim is being made - that a CAN model of 2D space needs to have a topology of 2D space (duh?)? Or that the connectivity in a CAN needs to be translationally invariant - which it doesn’t Clarke et al. 2025?
- The fact the model maybe gets multiple modules is awesome. Why was this not explored much more??? For starters, am I supposed to be able to see multiple modules from the ratemaps in figure 3 - that seems like an obvious thing to include, why is it missing? Do they think they are meaningfully different modules (as in Schaeffer et al. or Dorrell et al.) or related by discretisation effects as in Sorscher et al. (read end of page 7 of Sorscher et al. 2019 for discussion of illusions of meaningful modules)? Do you ever get more than two modules? How do they relate?
- 4.2.2 I think noise was just added to output neuron activity, why was this a useful control? Of course if you add not much noise to a grid cell it looks grid cell-ey, and eventually it doesn’t?
- I really disagreed with an argument in appendix C, line 748: ‘closer inspection revealed that the apparent success of the L2 capacity regularisation is not a general property, but rather a coincidence’ since L2 makes torus size reasonable. This is a very aggressive framing - why on earth shouldn’t I think that the success of the torus size regularisation shouldn’t be understood as an accident by which it approximates an L2 capacity constraint? They seemed symmetrically related, yet the authors argued theirs was better for reasons that completely eluded me.

**Questions:**

Many questions above.

---

### Official Review · Reviewer_VGnb · 2025-10-30

**Soundness:** 3
**Presentation:** 3
**Contribution:** 2
**Rating:** 6
**Confidence:** 4

**Summary:**

This paper proposed a theoretical framework to explain neural activity patterns in the brain, in particular during grid cells in the entorhinal cortex based on solving path integration tasks. Earlier computational work shows that (i) optimizing recurrent neural networks to solve path integration may lead to grid firing patterns; (ii) hand-crafted recurrent network that is capable of path integrating can explain the grid firing. There have been a substantial amount of work in this area to further investigate the origin of the grid responses. This work builds upon this literature and formulate a theory of grid cell responses based on three ingredients: task structure, representational structure and principle of isomorphism. In addition to the mathematical arguments, the authors present results based on numerical experiments. One main claim was that that conformal isometry alone is insufficient for the formation of the hexagonal grids.

**Strengths:**

***The paper is generally well written, and it is relatively easy to follow (although in various places, the descriptions were vague and some important assumptions were not fully justified).

***The authors developed a mathematical argument for the torus structure of the grid cells based on Gauss-Bonnet theorem under certain conditions.

***Additional numerical results were presented based on a feedforward, self-supervised network that preserves toroidal population activity. The authors explores the impact of different parameter values in the model.

**Weaknesses:**

***The paper focuses more on the mathematical formalism. The connection to the experimental data is weaker.

***The framework, while elegant, is incremental. Many of the ideas already appear in earlier work, e.g., Gao et al (2021), Xu et al (2022), Xu et al (2024), for example, some of the results in Section 3.2.

***Some of the findings were not carefully interpreted. For example, the abstract stated that the framework extends to 3-D grid cells. However, in fact the theory can’t explain the irregular structure of bat’s grid cells in 3-D.

***The numerical experiments were not directly linked to the theory. The authors showed that for large torus size, their network produce square grids. For small torus size, diffused grids were produced in their model. While these observations from numerical experiments were interesting, the paper would be stronger if the theory could explain why no hexagonal grids in these parameter regimes.

***There is a bit of a disconnection between the theory and the numerical results as well as biological realism. The theory appears to deal with path integration in an unbound space, but the numerical experiments were in a finite square environment. It is unclear whether having a bounded space would affect the theoretical results and proofs.

**Questions:**

What is the effect of the size of the arena? It is unclear whether the numerical results on the lack of hexagonal grid patterns for the extreme torus size was due to the (finite) size of the arena that was chosen. One way to get at this question would be very the size of the arena and see if the results reported in Fig. 4 would change.

Can the authors clarify whether their theoretical results would also work for bounded space? To me, Theorem 1 seems questionable for a bounded space, as a square in the representational space can also represent a square in the input space.

---

### Official Review · Reviewer_ReMv · 2025-11-01

**Soundness:** 2
**Presentation:** 2
**Contribution:** 2
**Rating:** 2
**Confidence:** 5

**Summary:**

The paper proposed Principle of Isomorphism (PIso), which preserves the essential structure of task-related computations, subject to biological constraints. The paper trains neural networks with an explicit torus embedding and generates hexagonal firing fields. The author interprets this as evidence that task-constrained, toroidal population structure is sufficient to produce grid-like single-cell patterns.

**Strengths:**

The paper is well resented and is in a good shape. The author is familiar with the related works, which brings a good foundation of this paper.

The paper makes a clear conceptual unification trying to show that 2 navigation tasks converge on the same topological principle.

**Weaknesses:**

1. I feel the paper is repacking the theory. The key theoretical step (compact, connected 2D Abelian Lie group-> 2-torus) is a standard assumption in grid-cell work and does not, by itself, add new structure or select hexagonal geometry.

2. The author seems to assume that emergence of grid cells is task-driven, but the model does not show that the same task objective makes multi-scale grid modules emerge on its own; instead, the torus size is effectively chosen or tightly controlled.

3. The paper questioned CI by grid-like fields only appear for an intermediate radius, which is more plausibly due to learnability/capacity/loss-weighting than to a failure of CI as a principle. It doesn't provide any explanation or analysis from the theoretical perspective of this behavior.

4. The paper doesn't theoretically strong, which means there is no direct proof or theoretically understanding of why the proposed method can lead to hexagonal patterns.

5. From the biological plausibility aspect, the loss function is not intuitive to me. I'm not sure if the brain is minimizing this specific loss.

6. The experiment results show that some neurons barely activate. For simulation, the environment is clean and simple, the patterns should be easily and perfectly learned according to previous paper. This makes me question the generality of the proposed theory. Also, the results are limited, which includes very little quantitative analysis.

**Questions:**

1. In my understanding, the torus size is related to the scale of patterns. The author makes a claim that CI couldn't lead to hexagonal patterns at certain scales, but Xu et al. (2024) shows that CI can learn different scales of grid cells with the change of a scaling factor. Can you explain more on this? I'm wondering if this "failure of CI" comes from learnability/capacity/loss-weighting issues.

2. In the experiment, the results are limited. To show generality, the authors need to make sure the principle can work under different scenarios, such as changing the network architecture/activation functions etc. Also, in the abstract, the author mentioned about HD system and 3D grid cells, but neither of them are studied in the experiment part. Could you provide more experimental details and more results to the paper?

3. The paper starts with the claim: because the two task that grid cells needs to achieve, the brain gets a torus and grids. However, this causal direction is not fully demonstrated. If the system is trained for path integration or metric reconstruction, why does the learning objectives not having the corresponding parts? And can author provide experimental results of these 2 tasks?

4. Moreover, I don't see much theoretical improvement or novelty since most of the assumptions and propositions are from previous works. Can author explain more about the genuinely new theoretical statement beyond re-deriving the torus from two task framings?

---

### Official Review · Reviewer_QGc8 · 2025-11-01

**Soundness:** 3
**Presentation:** 3
**Contribution:** 2
**Rating:** 2
**Confidence:** 3

**Summary:**

This paper proposes the Principle of Isomorphism (PIso), which states that neural population activity preserves the essential mathematical structure of the computational task it supports. Using grid cells as an example, the authors show that the two fundamental computations—Neural Metric (NM) and Path Integration (PI)—each independently imply a toroidal topology of population activity, derived respectively from flat Riemannian geometry and Abelian Lie group structure. The authors further unify both under Euclidean representation, and introduce a Topo-Constrained Network (TopoCN) enforcing toroidal topology to examine how single-neuron grid fields emerge. The model reveals that hexagonal grid patterns arise naturally within a specific range of the torus size parameter, while conformal isometry alone is insufficient. The paper concludes that representational topology, rather than local geometry, underlies the grid code and suggests “Topology Priors” as a general design principle for both brain and AI systems.

**Strengths:**

- The paper presents a clear and unified theoretical framework that links task geometry, group structure, and neural representation.
- The mathematical derivations connecting NM and PI to toroidal topology are concise and rigorous, offering an explanation consistent with empirical observations of toroidal population manifolds in grid cell recordings.
- The proposed TopoCN model provides a simple but effective platform to test the effects of topological constraints on representational structure.

**Weaknesses:**

- The theory remains high-level and abstract, with limited discussion of potential neural mechanisms that could enforce or approximate toroidal constraints in biological circuits.
- The emergence of hexagonal symmetry is shown empirically but not analytically derived.
- The torus-size parameter is empirically tuned and lacks a principled connection to biophysical parameters or network dynamics.
- Section 5 discusses distortion as an important feature distinguishing their approach from conformal isometry, yet the model itself does not demonstrate the ability to produce or analyze distortion in non-square environments, which weakens the connection between theory and empirical observations.
- The broader generalization of PIso beyond spatial domains (e.g., to conceptual or semantic spaces) is discussed but not concretely demonstrated.

**Questions:**

- Figure 3 shows that grid spacing appears inversely related to firing-field size—does this trend align with biological observations？
- In the loss formulation, the conformal isometry term employs a Gaussian kernel that effectively limits local sampling to an isotropic circular neighborhood. How sensitive are the results to this assumption? Specifically, if the kernel were made anisotropic (e.g., elliptical weighting or direction-dependent σ), would the resulting grid patterns exhibit systematic distortion or shear? Moreover, to what extent do the results in Figure 4 depend on the choice of the kernel width σ—does varying σ alter the stability or regularity of the emergent hexagonal grids?

---

### Note · Authors · 2025-11-26

I have read and agree with the venue's withdrawal policy on behalf of myself and my co-authors.